# Examining the challenges in sustaining user engagement with a mobile app to enhance multidrug-resistant tuberculosis (MDR-TB) care in Vietnam and its implications for implementing person-centred mHealth interventions

**Dorothy Drabarek**[1][◐], **Duy Trinh-Hoang**[2][*][◐], **Manisha Yapa**[1], **Tho T.H. Dang**[2], **Hai Dang Vu**[2], **Thu Anh Nguyen**[1,2], **Thu Thuong Do**[3], **Binh Hoa Nguyen**[3], **Dinh Hoa Vu**[4], **Greg J. Fox**[1,5], **Sarah Bernays**[1,6]

1 Faculty of Medicine and Health, Sydney School of Public Health, University of Sydney, Sydney, Australia, 2 Woolcock Institute of Medical Research, Ha Noi, Vietnam, 3 Vietnam Integrated Center for Tuberculosis and Respirology Research, Vietnam National Lung Hospital, Ha Noi, Vietnam, 4 National Centre of Drug information and Adverse Drug Reaction Monitoring, Hanoi University of Pharmacy, Ha Noi, Vietnam, 5 Faculty of Medicine and Health, University of Sydney, Sydney, Australia, 6 Department of Global Health and Development, London School of Hygiene and Tropical Medicine, London, United Kingdom

◐ These authors contributed equally.
* duy.trinhhoang@sydney.edu.au

## Abstract

Digital health technologies, especially mobile application (mHealth), offer great potential for enhancing person-centred care and managing MDR-TB. The rapid expansion of digital infrastructure in Vietnam presents a valuable opportunity for such interventions. This qualitative study examines user experiences to explain reported reduction in engagement with (i.e. use of) a smartphone app which is being trialed to improve early detection and management of adverse events (AEs) among MDR-TB patients in Vietnam (VSMART trial). We conducted 37 in-depth interviews with patients and healthcare workers (HCWs) and thematically analyzed the data. Initially, patients were motivated to use the app seeing it as a promise of the provision of 'good care' from trusted healthcare workers, and overestimated its functional capacity. However, as patients realized its functional limitations for AE reporting and management they reverted to communicating with HCWs through existing communication channels. While the app empowered patients to communicate with HCWs for AE reporting, it inadvertently increased HCWs' workloads which became difficult to manage. This resulted in a paradox where the app was used because of its social value, in spite of its limited functional value. This study reveals how relational and socially mediated effects of technology may complicate mHealth design and implementation, illuminating why 'acceptable' technologies within pilot/projects could struggle to sustain engagement a scale. To support attention to this process, we propose an explanatory framework that captures the social-functional dynamics of mHealth interventions which can guide much needed qualitative evaluation to support the design of mHealth

**Data availability statement:** The data used in our study is highly identifiable and sensitive; therefore, making it publicly available would violate the terms of the informed consent provided by participants. Our study protocol (number 2019/676), approved by The University of Sydney Human Research Ethics Committee (HREC) and Vietnam National Lung Hospital Institutional Review Board (number 13/19/CT-HDDD), explicitly states that data will only be shared upon reasonable request in a manner that ensures participant confidentiality, except where required by law. Interested and qualified parties may submit data requests to the following sources: In Australia Manager of Research Ethics, University of Sydney: • Phone: +61 2 8627 8176 • Email: human.ethics@ sydney.edu.au • Fax: +61 2 8627 8177 In Vietnam (local ethic approval), Dr. Nguyen Viet Hai Secretary of the Ethics Committee, Vietnam National Lung Hospital • Phone: +84 98 201 82 22 • Email: nguyenviethai.hmu@gmail.com

**Funding:** This research is funded by Vietnam National Foundation for Science and Technology Development (NAFOSTED) under grant number NHMRC.108.02-2018.0. The funders had no role in study design, data collection and analysis, decision to publish, or preparation of the manuscript.

**Competing interests:** I have read the journal's policy and the authors of this manuscript have the following competing interests: One of the authors, Thu Anh Nguyen, is an editor at the PLOS Global Public Health journal. The remaining authors have no conflicts of interest to declare.

technologies to align with stakeholders' needs and suited for integration into local healthcare systems.

## Introduction

Inequity characterises and drives the global distribution of morbidity and mortality associated with tuberculosis (TB) and the resources and socioeconomic development necessary to eliminate it [1–3]. Building on increasingly widespread use of mobile digital information and communication technologies (ICT) in Vietnam, mobile health (mHealth) is championed as an innovative sociotechnical approach that can overcome some of the structural conditions which impede the effectiveness and inclusivity of TB care [4] and shape inequities in TB outcomes [5].

In Vietnam, as in other resource-stretched settings, multidrug-resistant tuberculosis (MDR-TB) is presented as a significant barrier to achieving the World Health Organization's (WHO) End TB strategy goal of a 90% reduction in TB-related deaths by 2030 [6]. The drug regimens used to treat MDR-TB cause a range of mild to more serious adverse events (AEs) and require close monitoring to avoid harm to patients. Severe AEs occur commonly, even among patients without comorbidities, and often lead to treatment interruption [7] and poorer or failed treatment outcomes [8].

Developing mHealth interventions to support early detection and management of AEs [9], has become a key priority of MDR-TB care [10]. An influential appeal of mHealth interventions for MDR-TB is the expectation that they offer a scalable means to deliver more patient-centered care [9] where monitoring, which can be done at a distance, is provided through a support rather than a compliance or directly-observed model [11,12]. The mHealth technologies that may fulfil this vital function include text messaging, video calls, and electronic medication reminders, as well as apps to improve adherence support and monitoring, diagnosis, individualized dosing, eLearning, and self-testing [13,14]. However, despite the high expectations, there is limited evidence for the effectiveness of such mHealth interventions for improving patient-important outcomes [14,15] and on the monitoring and reporting of AEs [16]. Similarly across a range of communicable and non-communicable conditions, there is a limited and mixed evidence-base demonstrating the 'effectiveness' of mHealth technologies in improving treatment outcomes and healthcare delivery in practice [17–19], or their positive impact on health systems [20,21].

An emerging pattern within mHealth research is that although many mHealth technologies have initially seemed promising, i.e. have been perceived as 'acceptable', once they are moved from the controlled environments of trial research into being implemented, their sustained use and scale-up is often disappointing [17,22–25]. This phenomenon is referred to as "pilotitis" [26,27]. Acceptability (domains of which include: affective attitudes, intentional use, and pre- and post-use satisfaction) has tended to be considered an initial predictor of mHealth success but empirical studies indicate that it is commonly not sustained [23,28]. Digital health researchers call for a more granular exploration of related but distinct concepts of engagement, encompassing behavioral (usage patterns), cognitive (interest) and affective (enjoyment) dimensions to develop a more nuanced understanding of the variation in what underpins initial versus sustained user interaction when considering mHealth intervention effectiveness and widespread adoption [25,29].

To adequately understand engagement it is necessary to move beyond an approach focused on individuals to examine the dynamic interaction between human action, novel interventions and the systems into which they are implemented [23]. This requires a shift in the

dominant conceptualisation of mHealth interventions within the literature from considering them as discrete, immutable entities, serving functional purposes for individual users [30] to adopting a broader systems-focused lens. A wider systems-focused lens is increasingly being adopted to demonstrate that the use and meaning of technologies evolve and take shape in practice and context as users co-opt, and adapt them in ways that make sense to them but may diverge from their intended use or design [26,31,32].

Integrating qualitative research in the evaluation of mHealth interventions [33] is key to explicating the contextual and relational influences shaping users' engagement within a specific health system. We present a qualitative evaluation of users' experiences of an mHealth app intervention that aimed to improve early detection and management of AEs within the 'Harnessing New mHealth Technologies to Strengthen the Management of Multidrug-Resistant Tuberculosis in Vietnam' (V-SMART) trial. This intervention was aimed at encouraging a more patient-centered approach to delivering MDR-TB care.

This qualitative study sought to explain why although there was initially high uptake, there was an unexpected decline in app engagement (use), with the proportion of patient users checking in at least 5 days/week dropping from 58.6% in the first month (October 2020) to 8.3% by the eleventh month (forthcoming publication).

## The V-SMART trial and the 'Bac Sy Minh' app

The V-SMART trial being conducted in urban and rural settings of seven provinces in Vietnam (both rural and urban) between 2019 and 2025. The mHealth intervention being tested is the 'Bac Sy Minh' (which translates to 'Dr Smart' in English; Minh is also a Vietnamese first name) smartphone app. It has both patient-facing and HCW-facing versions. Its aim is to encourage patients receiving standardized treatment for MDR-TB (typically a 9- or 20-month treatment regime containing a minimum of five antibiotics) to report potentially concerning changes in how they were feeling in real time to healthcare workers (HCWs), and to make this process more equally accessible among patients. Designed as a triage tool, it was intended that the app would identify and prompt further investigation by HCWs to support the timely and effective management of potential AEs. The VSMART trial protocol contains further information and rationale for the design and implementation of the intervention [34].

Patients were invited to participate within the first 90 days of commencing treatment at a government programmatic management of drug-resistant TB clinic, either while an inpatient or upon discharge from hospital. The control group received routine MDR-TB care without the smartphone app. The primary trial outcome was improvement in WHO-defined treatment outcomes among patients treated for MDR-TB [35]. Participants are followed up for 24 months and in both arms, receiving care for MDR-TB that is delivered through community, district and/or provincial level health facilities.

Trial participants - patients and HCWs - are requested to register an app account using their phone number (as unique identifier). Those randomized into the intervention arm attend a group training session run by study staff on how to use the app along with videos and a printed manual. They are paired (via their app account) with a HCW from within their province or district, who is (preferably) their primary HCW for MDR-TB care and receives and responds to their activity within the app. Provided their smartphone is connected to the internet, the app provides a daily alert to patients to remind them to take their treatment each day at the same time. If the patient engages with the reminder, they proceed to a sequence of prompts to determine the presence of somatic or emotional symptoms and/or treatment side effects. The assigned HCW is then notified via the app if any concerning symptoms are identified and received weekly reports via the app identifying patients with suboptimal adherence.

The patient- and HCW-facing version of the app also included a direct messaging feature, and the patient-facing app included a 'frequently asked questions' section containing MRD-TB literacy materials and routine culture results.

Given that participation relied on access to a smartphone, participants (both patients and HCWs) receive a small monthly data allowance (70 000 VDN/approximately 4.40 AUD) which was paid as a reimbursement. A smartphone was loaned to sixteen intervention arm participants (3.56%) who did not have their own or whose smart phone was not compatible with the app, one of whom participated in the qualitative study.

The mobile app was developed for iOS and Android operating systems and was designed in consultation with HCWs, patients, ICT technicians, policymakers, researchers, and software developers. A pilot evaluation, including in-depth interviews with HCWs and patients who tested the app was conducted prior to the commencement of the trial. The findings of this initial analysis will be published separately.

## Methods

### Ethics statement

Ethical approval for the V-SMART trial and this qualitative sub-study was granted by the Human Research Ethics Committee of the University of Sydney (2019/676) and the National Lung Hospital (971/QD-BVPTU), Vietnam. We obtained written consent from participants.

### Design

Qualitative research was conducted with patients and HCWs who were using the app during the early implementation phase of the V-SMART trial (October 2020 to December 2022). We used individual in-depth interviews to understand the variance in patterns of use, the app's acceptability and its perceived value from the users' perspectives.

### Sampling

Adopting a purposive sampling approach, we aimed to recruit participants with maximum variation to reflect the diversity of the trial population. In one northern and two southern provinces where participants were most actively being recruited into the trial we selected patient participants from those who had indicated to their HCWs that they were interested in the qualitative study. Our selection was informed by the key sampling variables of sex, age, and urban/peri-urban/rural residence, within this we also sought to include trial participants whose app usage reflected the two dominant patterns identified within the quantitative trial data: 1) those who had stopped using the app after a period of consistent engagement including those who used the app once or only a few times upon enrolment, and 2) those who used the app sporadically across their treatment. Sampling concluded when we had reached thematic saturation and when we were satisfied that the distribution of participant characteristics across the conceptual framework reflected established patterns of risk factors for TB [6] and expected associations between patient characteristics and use of novel technologies. For HCWs, we sought to reflect the diversity of health facility locations (urban and rural) within the trial, prioritizing HCWs who were paired with patients whom we had already interviewed.

### Recruitment and data collection

Participants were recruited between November 2021 and August 2022. Patients were introduced to the qualitative sub-study by their treating doctors. Interested patients were contacted

by phone by a Vietnamese qualitative researcher (DT, male) to discuss the qualitative study further. The primary qualitative researchers working on this study (SB, DD and DT) were not members of VSMART trial team and did not have prior relationships with potential participants. HCWs were also contacted by the research team to discuss the study and were invited to participate.

In line with COVID-19 restrictions at the time of data collection, most interviews were conducted remotely by telephone or Zalo call (a Vietnamese social media platform that supports digital communication via phone call, text, voice message, photo and video). Participants received a digital link to the participant information sheet (PIS) and completed a consent form via their smartphone. Written consent was obtained for the seven participants who were unable to provide consent digitally.

We selected and approached 28 patients. Three declined due to scheduling conflicts. The 12 HCWs participating in the trial in the three provinces selected as qualitative sites agreed to participate. Individual in-depth interviews were conducted by DT, either via phone (n = 18), Zalo (n = 12) or in-person (n = 4). Once local COVID-19 restrictions were lifted, further interviews were conducted in-person (n = 3) to ensure the data collection method was not inadvertently excluding the experiences of those who were not comfortable engaging with the app and/or the study via phone [36]. Adapting our study to incorporate remote data collection while maintaining opportunities for in-person participation allowed us to explore variability across participants' personal digital literacy and confidence in engaging with a mobile app [37]. Interviews were conducted at a mutually convenient time and place by an experienced Vietnamese qualitative researcher, were audio-recorded with the participants' permission and lasted between 45–90 minutes.

Semi-structured topic guides were designed to be used flexibly for interviews with patients and HCWs. The interviews with patients focused on their experiences of illness and treatment, support and communication using digital technologies, using the Bac Sy Minh app and its implementation, and potential approaches to increasing its use. Interviews with HCWs focused on their experiences of using the app, including to what extent the app supported MDR-TB care delivery. In later interviews we discussed our emerging interpretations, as a form of validation. These were also presented to trial staff to relay back the implementation challenges they encountered by patients and HCWs.

## Data analysis

Thematic analysis was conducted, an approach well suited to understanding nuanced issues surrounding use and implementation of mHealth [33]. This began with the writing of a detailed summary and field note after each interview. The qualitative research team met after every 2–3 interviews to discuss the interview summaries in-depth, which informed both a refined interview guide and further sampling decision [38]. These discussions were written up into analytical memos to track early analytical ideas.

All audio recordings were transcribed verbatim, with identifying details removed. A subset of interviews, were selected for their rich accounts of topics of interest and were translated into English for the non-Vietnamese-speaking researchers. DD and DT collaboratively developed preliminary codes from interview summaries. Once the analysis team reached a consensus, the codes were applied to the full transcripts using Microsoft Word. They were progressively refined and organized, within Microsoft Excel, into key themes [39] linkages between patterns of app use, participant characteristics, and key themes increasingly honed. This iterative process of interpretation and conceptualization produced a conceptual framework inductively [40] to explain the observed reduction of app use among patients, which we present within an overall explanatory narrative [41].

## Results

### Sample

We conducted a total of 37 in-depth interviews with 25 patients and 12 HCWs whose characteristics are summarized in Table 1. The higher number of male patients recruited for the study reflects the disproportionate impact of TB on the male population [6]. HCWs were doctors, nurses and public health officers working in district and provincial level TB clinics.

Although our results are presented sequentially, Fig 1 illustrates inter-related nature of the results. We first describe the pre-existing standard of care for managing AEs to characterize the 'relational care space' into which the mHealth intervention was introduced. We then examine the experiences of using it from the perspective of HCWs, and patients to identify the key moderating factors shaping the app's uptake and use. We unpack the consequences, including those that were unintended, of the app in the final section.

### Before the introduction of the app: management of AEs using local logics of patient-centered care

The majority of patient-participants reported experiencing side effects from MDR-TB medication, HCWs described these as occurring most commonly early in treatment. Patients were monitored in hospital for two weeks immediately after diagnosis. Once discharged, they then attended routine outpatient care. The formal process for which involved: 1) daily patient visits to a local commune TB clinic for Directly Observed Therapy (DOT) and monthly visits to a district or provincial level TB clinic, 2) *ad-hoc* communication via phone with commune HCWs to report AEs, which would be triaged and treated within the commune TB clinic or 3) complex cases were referred to the district or provincial TB clinic. In practice though patients adapted these arrangements, in negotiation with HCWs, to fit their needs and personal circumstances.

'Quan tâm' encapsulates how patient-participants consistently described their care. This Vietnamese phrase captures the attentive healthcare provided by HCWs. Patients described that they display genuine interest in them as individuals, commensurate with definitions of 'person-centred care'. We interpret their sentiment here as 'good care'. A female patient gives an example of how this was demonstrated in practice: "What I like, is that the hospital keeps a close eye on me." (33, urban Can Tho, tour guide). Patients felt respected and valued. Female HCWs, in particular were praised for providing counseling that considered how TB treatment could be integrated effectively into a patients' life circumstances.

Table 1. Participant characteristics (patients and HCWs).

| Characteristic | | Patients n (%) | Healthcare workers n (%) |
|---|---|---|---|
| **Total** | | 25 (100%) | 12 (100%) |
| **Province/ City** | An Giang | 10 (40%) | 5 (42%) |
| | Can Tho | 7 (28%) | 2 (17%) |
| | Ha Noi | 8 (32%) | 3 (25%) |
| | Ho Chi Minh City | - | 2 (16%) |
| **Sex** | Female | 8 (32%) | 8 (67%) |
| | Male | 17 (68%) | 4 (33%) |
| **Age (years, range)** | | 24–63 | 28–54 |

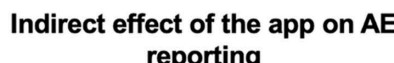

**Fig 1. Organization and inter-related nature of the results.**

*"Dr X [female] gave me so much, not only medical attention but also encouragement regarding personal issues, so I got through the difficult time. One time I coughed up so much blood as I was angry at [personal issue], X helped me not only medically but also emotionally to power through that dark time to continue with treatment otherwise I would have given up…I am so grateful for her…Now whenever I feel uncomfortable, I can just call X"* (Female patient, 55, rural Can Tho, farmer).

Similar characteristics of the provision of person-centred care was described by HWs. They explained how the care they provided was based on their intimate knowledge of patients' needs, attention to how their broader circumstances were likely to influence their engagement with treatment, and a deep commitment to supporting the well-being of their patients.

*"For good AE management, it is important to reduce the 'distance' between doctors and patients, not a physical but a social one. It is about trust. If patients don't trust me, they won't tell me about their symptoms."* (Male doctor, provincial level, Ha Noi).

The emphasis on 'good care' shaped the processes of AE reporting and outpatient care in practice. Many patients tended to report AEs directly to the provincial TB clinic when they considered it necessary. Others described how their requests for changes in where and how frequently they received their medication were acted upon.

*"I was told to go to the district TB clinic every day for medication and monthly tests. But I would like go to the Hanoi Lung Hospital, so I asked them, and they agreed. Now I collect a month's worth of medication on the day I visit the hospital to get tested."* (Female patient, 50, rural Hanoi, informal labor).

Patients who were identified by HCWs as 'at risk' of developing AEs and/or suboptimal adherence and/or discontinuation of treatment were supported to have more direct access to a provincial-level doctor who had the clinical authority to make changes to their treatment regimen or prescribe supplements to manage AEs. HCWs also shared personal mobile numbers and were willing to connect with patients over Zalo, the most widely used communication app in Vietnam. This enabled patients to circumvent the formal hierarchical system of reporting and decision-making (from commune to district to provincial level and back). HCWs justified these practices by emphasizing how direct communication served to strengthen and sustain patients' trust, as well as their confidence in reporting AEs and engagement with treatment during periods of severe side effects. This strategy, however, created unequal access to HCWs which may have indirectly contributed to uneven clinical AE monitoring.

## Working with the app: limited and tentative use among HCWs

The Bac Sy Minh app was introduced into the local organization of MDR-TB care delivery to enhance existing detection and management of AEs. HCWs explained that their use of the app was contingent upon how well it enabled the delivery of patient-centered care and to what extent it assisted the existing organization of their work. Recognizing its potential benefits, HCWs detailed aspects of the app which they felt undermined its acceptability.

**Unmet expectations: the app's limited functionality to engage with existing approaches to delivering person-centred care.** HCWs expected that a new technology would streamline their work by addressing health system challenges and supporting how MDR-TB care was locally delivered. In many cases however, these expectations often exceeded the app's intended scope. Firstly, HCWs explained that some of the medical problems that patients experienced were related to co-morbidities or adverse personal circumstances. For many HCWs, the inclusion of only predefined MDR-TB related symptoms and treatment side effects was too narrow to capture such complexities. HCWs also described how patients struggled to align their embodied experiences with the descriptions of AE symptoms in the app. Female HCWs, in particular considered that the app, which relied on an algorithm, could not adequately provide the emotional support necessary to sustain patients' engagement with treatment during difficult periods: *"But I cannot comfort patients with the app!"* (Female nurse, provincial level, Hanoi). HCWs expressed a preference for the direct dialogue facilitated by a phone call to identify complex clinical symptoms, explain relevant information, and provide the necessary responsive care. This highlights a disconnect between the assumed function and intent of the design. The app was designed as a triaging tool to highlight the need for further person-centred action, for example to alert HCWs to the need for direct discussion with their patient. HCWs' dissatisfaction may stem from a misunderstanding about the remit of the app.

**Misalignment with local MDR-TB system solutions and challenges.** Further, HCWs described that the way the app functioned did not align with the local organization of AE

monitoring and response. In a three-tiered TB care system (commune, district, provincial) facing resource constraints, HCWs had adopted a task-sharing strategy to manage workloads. Within MDR-TB units and across levels of care management, available staff would respond to patients' requests when others were occupied. The app though, was designed to facilitate dyadic communication between patients and assigned HCWs, primarily at the commune and district levels. However, as patients commonly bypassed this arrangement and sought care from senior doctors at provincial level, an existing uneven distribution of patient-HCW communication persisted and the app could not assist in shifting care back to district and commune levels. The app did not lend itself well to integrating into this flexible system of horizontal and vertical task-sharing.

> "We don't have many staff, so we share our work within our hospital and with HCWs at district and commune levels. The app does not allow us to do that, it was all on me." (Female doctor, provincial level, Can Tho).

HCWs appreciated the app's innovative intent but found it challenging to integrate such an 'advanced' discrete technology into Vietnam's fledgling digital health infrastructure. Given the lack of connectivity with existing IT and communication systems, the app operated in isolation, duplicating tasks rather than streamlining them. HCWs expressed a desire for connectivity between the app and other digital systems to realize its potential to enhance existing delivery of person-centred care.

> "There is hospital management software that processes large amounts of patient data. They alert cases that need immediate attention, just like the app. That's what we need, but those packages are too expensive to purchase. Why not help us invest in such a digital system instead so that the app can connect with our data system, otherwise the app is just an add-on that creates more work for us." (Male doctor, provincial level, Hanoi).

## App uptake and use among patients

Patients' engagement with the app tended to taper after the initial weeks of use. They commonly moved to other communication platforms, namely Zalo, to report AEs to their HCWs. We explain why below.

**Initial enthusiasm: how good care facilitated intervention uptake.** The majority of patients agreed to use the app because it was recommended by their HCW. Patients interpreted the invitation as a symbol of good care, representing HCWs' investment in supporting them to achieve optimal treatment outcomes. As an act of reciprocity, agreeing to use the app served as an opportunity to demonstrate being a 'good patient'. For many patients, the app assumed a strong social value, symbolizing the good care being provided and the therapeutic relationship.

> "I know it was voluntary, but I wanted to participate. The doctors want good things for their patients, this is why they asked me to use the app." (Male patient, 24, rural An Giang, unemployed).

The enthusiasm to enroll in the trial and use the app was thus entangled within the existing care relationships between patients and their treating HCW. This is illustrated by how some patients named the app after their HCW, for example "Dr Y's app." (Male patient, 49, rural An Giang, street vendor).

**Evolving app use.** Initially encouraged by HCWs, patients experimented with using the app in various ways. The practices and logics underpinning patients' app use converged into four patterns. The delineation of which was based on the complex interplay between varying degrees of perceived *social value*, a product of how the app symbolised good care, and the *functional value* expected from the app (Fig 2).

1. Functional use

Within the qualitative sample only a small number of patients cited the app's functionality as their primary reason for using the Bac Sy Minh app. These patients' use of the app, however, was short-lived because it did not meet their expectations. This could be in part due to misunderstandings about what the app aimed to achieve. Upon realizing that the app did not perform the functions that they had anticipated well, exemplified in delayed update of culture test results or medication reminders that were not useful for them, they stopped using the app and reverted to pre-existing mechanisms to communicate, which they described as more familiar and appropriate to meet their needs in reporting AEs and discussing their concerns with HCWs.

*"So, I was most looking forward to automatically receiving the [microbiological] test result monthly, but as the information was not sent to me, I deleted the app… About reporting side effects, this function is not too important for me because if I wanted to ask anything I can just call my doctors, I have their numbers which I asked for."* (Female patient, 32, urban Hanoi, accountant).

Patients in this group tended to be young (in their 20s and 30s), of high socioeconomic status and were comfortable using digital technologies. For these patients, confidence that they could navigate direct communication with HCWs through other channels diluted the social value of the app, in terms of its relational benefit, because they were not concerned that disengaging from it would adversely affect their relationship with HCWs.

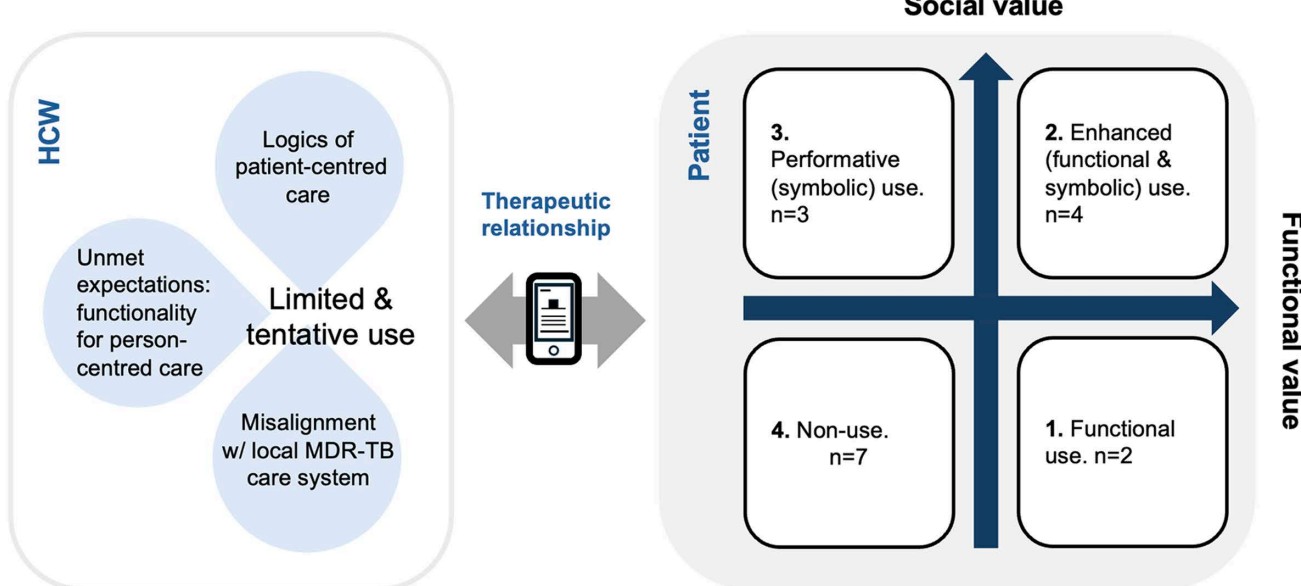

**Fig 2. Conceptual framework of practices and logics underlying four patterns of patients' app use.**

2. Enhanced (functional and symbolic) use

For a small number of patient users, the app's utility was based on both its functional and social value. This group considered the app to have functional utility for facilitating AE reporting and their use of the app aligned with how it was intended to be used - they used the daily check-in, the symptom algorithm and the app's texting feature to communicate with their HCW. The patients in this group tended to be younger (in their 20s) and were proficient in using digital technologies. However, unlike the first group (functional use), patients in this group considered themselves to have limited agency as patients and were subsequently less confident to contact HCWs at their own discretion outside of the app. The app therefore operated as a permissible, novel mechanism through which they could contact their HCW.

*"I was too shy to talk to him [Dr] as I think he is often busy… I often updated my health status on the app, and sometimes I notified the doctor that I was not feeling good by texting directly on the app, and then the doctor would reply to me."* (Male patient, 24, rural An Giang, kitchen assistant).

3. Performative (symbolic) use

The majority of patient users in the qualitative study sample were concentrated within this third category. It included those whose app use declined gradually until they either no longer used it or only sporadically checked in (as captured in the trial's quantitative monitoring data). They gave positive accounts of the app, but their use tended to be more performative than functional, using it because they valued the 'good care' it represented and used it as an opportunity to actively demonstrate being a 'good patient' who was deserving of 'good care'. However, over time, their communication with HCWs about AEs migrated out of the app to other platforms that they found to be more functionally effective.

A 49-year-old male patient from rural An Giang use of the app was broadly representative of how most patients reported engaging with the app. He frequently reported that he was 'Feeling very good!' through the app, despite at times feeling some discomfort. In explaining this disparity he said that, "*I just do what doctors told me, I open the app and click 'Feeling good'*". This exemplifies the practice of 'performative use', engaging in its social value and using the app as a means to demonstrate to his HCW that he was using the tool they had given him, rather than as a way to accurately report how he was feeling. This group encompassed a broad spectrum of patients whose app engagement was motivated by the opportunity to demonstrate compliance. Although performative users progressively reverted to other direct modes of communication, this occurred in parallel to using the app, which ultimately declined over their treatment period. Even when trial follow up overlapped with periods of COVID-pandemic control measures in Vietnam, app use did not increase for this majority group, despite the expectation by its implementers that it would help to accelerate the transition to digital TB care. Given that this category of patients reflects the dominant pattern of usage, it highlights that high uptake and some form of engagement, albeit dwindling, during the initial period of treatment is not a reliable indicator of longer-term use and adoption.

4. Non-use

There was a fourth group of patients, a substantial minority, whose inability to use smart-phone apps precluded them from engaging directly with the app. In line with the guidance provided, these patients sought the support of family members to use the app.

*"I don't know what this app does, I just do what the doctor asks me to do. My grandson pressed buttons on the app for me. But now he come back to his mom's [her daughter's house]"* (Female patient, 55, rural Can Tho, farmer).

Their use was minimal to none, in part because their family members were not available to consistently facilitate engagement with the app. These patients tended to only passively engage in MDR-TB care, and had particular characteristics that HCWs identified as markers of risk and in need of more intensive AEs monitoring. These included being elderly often with co-morbidities, having less access to material resources, and having a secondary-school education or less. HCWs anticipated that this group of patients would struggle to engage with the app and continued to pro-actively provide alternative means of communication for AE reporting that were more suitable and accessible, such as offering contact through their personal phone numbers.

### Indirect effect of the app on AE reporting

Overall, despite patterns of declining or limited use of the app, HCWs described an increase in the reporting of AEs, as well as overall communication with patients. This was not necessarily via the app, rather via phone call, text, Zalo, or Facebook. As a symbol of care, the app provoked an influx of communication. For many HCWs this exceeded their capacity to respond. This became a challenging burden, particularly for provincial level and female HCWs. They described that the app made it more difficult for them to manage patients' expectations around what constituted an appropriate frequency of contact. Using methods of communication other than the app had allowed provincial level HCWs to prioritize their attention towards those who they deemed 'at risk'. But others appreciated that the triaging function of the app had some capacity to discern between those who needed urgent help and those who could wait. It was in part enabled by a flaw we identified in the app's design that HCWs' phone numbers were visible to patients.

*"I only give my personal number to patients who tend to have more problems with medication. But now most of the patients [with the app] can access my number, they now text me too, I have to answer… it is good to help patients contact the senior doctor directly, but it becomes too much for me to handle."* (Male doctor, provincial level, An Giang).

Female HCWs were most affected by the increased workloads. They described how patients often contacted them to seek emotional support that was sometimes unrelated to their illness or treatment. This led them to exercise discretion about whom they provided their personal contact to. Once the app was introduced, this became more difficult to do. For many patients, the app implied the constant availability of HCWs and permission to access good care at any time.

*"As the only treating doctor here, I am strategic with giving out my [personal] phone number to patients. I only give it to those most at risk of AEs... Others can just call the landline phone of our TB unit. Otherwise, I would be inundated with unnecessary phone calls and messages. But since using the app I have become more of a mental health counsellor as I am now presented [unwittingly] as available to discuss personal issues too."* (Female doctor, district level, Ho Chi Minh City).

## Discussion

We qualitatively investigated the reduction in user engagement with an mHealth app being trialed to enhance the management of AEs within routine MDR-TB care in Vietnam. For patient users, the app had a social value through which good care could be demonstrated and negotiated between patients and HCWs. Desire to engage in this interaction drove its initial

uptake among patients. This increased AE reporting and the perceived universal accessibility of HCWs, even for those less likely to seek contact outside appointments. However, over time, communication tended to migrate out of the app because it did not have superior functional value for AE reporting in practice compared to alternative communication channels. Increased communication about AEs occurred largely *outside of the app*. The dual nature of which, symbolically and functionally, led to a paradoxical phenomenon: the app increased HCWs' workload, causing dissatisfaction, and while patients liked and appreciated it (high acceptability), it was not used as intended or as often as expected. Despite its catalytic effect the app's social value alone was not enough to sustain users' engagement as its intended functional value needed refinement. This study highlights insights provided by qualitative inquiry in illuminating the drivers of use and pathways influencing behaviors like timely AE reporting, which may not appear in trial data.

## Lessons for mHealth interventions for MDR-TB care

The findings provide valuable insights for improving mHealth for MDR-TB care. Understanding of AEs may vary (42). In the case of the Bac Sy Minh app, AE management was complicated by differing perceptions of AEs between patients, guidelines, and the app. Predetermined descriptions may not align with how patients report experiences, highlighting the need for flexible AE classifications to better reflect patient priorities and improve person-centred care.

Reliance on pre-existing AE reporting methods such as Zalo and other social media reflects "informal mHealth" that predates the app [42]. Leveraging these practices while addressing HCWs' unmet expectations can create more inclusive communication platforms. Effective intervention design requires understanding local care contexts and empirically assessing impacts, over time [43].

mHealth tools must address user challenges and health system complexities. HCWs' unmet expectations reflect broader health system issues that, inevitably, no single app can resolve [5,44]. Sustained mHealth impact requires intersectoral engagement to avoid ecosystem fragmentation [44–46] and to support addressing contextual challenges to prevent overburdening staff, undermining confidence, and discouraging engagement [24,47].

## Contribution to conceptualizing the use of mHealth technologies

Our findings show how the app acquired meaning and possibilities for end users [48]. This resonates with the emerging literature which highlights the symbiotic way in which the use of technologies is influenced by how "people engage in affective relations with and through technologies" [32] and how symbolic connection can shape and alter care practice [49]. Our analysis highlights that the role of social value was more influential in shaping initial uptake and use of the app than the promise of its functionality, which we argue may explain the limited engagement. While specific to its context, this mirrors findings from an SMS-based TB treatment intervention in Burkina Faso, which also demonstrated social value but had limited functionality [30].

Our focus on engagement and use patterns highlighted some valuable dimensions for explaining how acceptability mediates effectiveness [25]. Recent research shows how novel digital technologies often do not function as anticipated when implemented into complex healthcare environments and systems [23,50]. Ensuring that we adopt a temporal lens enables us to observe critical changes, such as fluctuating uptake and use, over time [25,28]. The dislocation that we noted between intended use as designed, and the anticipated functionality among users highlights the criticality of attending to the intervention's coherence (i.e., how well participants comprehend the intervention and its mechanisms) as defined within Sekhon

et al.'s acceptability framework [51] in understanding how initial enthusiasm may not translate into engagement, sustained use and effectiveness for users.

The social value users ascribe to mHealth technologies, therefore, may explain why those which seem to "work" within pilots may not translate as well when implemented at scale and integrated into routine care [22–24,26,27,52]. We argue that symbolic influence may be a critical factor that complicates or distorts the assessment of an mHealth intervention's acceptability and utility, and has implications for interpreting effectiveness. The mHealth symbolism manifested in enthusiastic uptake, such as the influence of the social value of the app that we observed in this evaluation, may artificially imply effectiveness while concealing limited functional value. Adopting a relational lens to understanding intervention effect is critical to guide mHealth developers and implementers in anticipating and responding to social relationships that are formed and moderated through digital technologies, and the distorting effect they may have for sustainable implementation. Accounting for social relationship dynamics among different user groups helps identify implementation risks which are often overlooked in evaluations focusing on a single type of user. While we do not diminish the contributions that interventions which convey social value may have, to achieve their intended effect mHealth interventions should be transitioning users toward functional or enhanced use, while shifting them away from performative or unproductive use patterns.

**Integrated participatory approach in patient-centred mHealth intervention.** The app's design employs a human-centered approach as a means to develop mHealth technologies that support patient-centered MDR-TB care [53]. There is a risk that this design approach may inadvertently be applied with too narrow a focus on individual user needs and adopt a cross-sectional assessment. This may compromise focus on the complexity of evolving doctor-patient dynamics and the broader healthcare system over time, both of which are important considerations for enhancing patient-centered care [28]. The flexibility to incorporate iterative learning which may emerge over time is key to ensuring that an intervention's design can accommodate the relational and systemic context of implementation [54–57]. The V-SMART trial is an example of incorporating an iterative evaluation process that extends beyond the initial design and pilot stages [58,59]. This is a key component underpinning the stepwise or incremental design and implementation process of mHealth technologies, as suggested by Peiris et al., to avoid the so-called 'killer app' which tends to overengineer and overstate the value of digital mHealth solutions, while de-emphasizing user needs and contextual factors [60,61].

## Strengths and limitations

This study examines technology-mediated interactions between patients and HCWs, an underexplored topic within mHealth literature [56,62]. A qualitative approach can identify important unintended consequences of this mHeath technology, which would otherwise not necessarily be detected within trial data. Triangulating data from both groups also helped mitigate self-reporting biases.

We also developed a novel conceptual framework that accounts for social drivers of engagement with mHealth technology. Initially our sampling approach focused on recruiting from within two broad groups, higher use participants and low use participants. Within our iterative analysis we identified four explanatory categories of use, which were unevenly distributed across the sample. We oversampled for higher use in an attempt to include more participants displaying functional and enhanced use. However, they remained rare (e.g. only two participants in the functional use category). Even recruiting for specific sampling characteristics that we tentatively considered may be an indicator of these categories, for example high technological literacy, youth, and higher socioeconomic status, did not lead to further identification.

With limited study resources, recruitment concluded once we reached thematic saturation, both in terms of consistency within the categories and no further categories of use emerging. The distribution of the sample across the categories, to the best of our knowledge, reflects the broader patterns identified in usage within the trial participants at the time of the qualitative study being conducted. Given that our findings are shaped by the context of this study and the cross-sectional enquiry, we welcome further investigation into its transferability within and beyond TB-focused AE notification apps and the Vietnamese context.

The study's limitations highlight areas for future research. Including perspectives from those involved in the app's design and development could expand findings to better capture mHealth development dynamics. The financial impact on app use is under-explored. Future research should examine the role of financial incentives on app engagement. This study only investigated the early stage of implementation. A mixed-method, and longitudinal design should be considered to extend the interpretation of the findings.

## Conclusion

This qualitative study of mHealth in MDR-TB care in Vietnam reveals the significance of social influence on engagement with digital health technologies. The social value ascribed to the intervention technologies, which was not sustained in light of experienced limited functionality, may provide some explanatory insight into the challenges facing many mHealth interventions. We advocate for an iterative approach, utilizing qualitative inquiry, to development of patient-centered mHealth interventions adaptive to technology-moderated therapeutic dynamics, the local needs and priorities of end-users, as well as the complex organization of care provision in practice.

## Supporting information

**S1 Checklist. Inclusivity in global research.**
(DOCX)

## Acknowledgments

We are grateful to the study participants for dedicating their time and sharing their experiences with us. We also extend our thanks to the colleagues in the V-SMART research program and the local partners' staff at district community health centers in An Giang, Can Tho, Ha Noi, and Ho Chi Minh for their valuable assistance in recruitment.

## Author contributions

**Conceptualization:** Dorothy Drabarek, Tho TH Dang, Thu Anh Nguyen, Binh Hoa Nguyen, Dinh Hoa Vu, Greg J. Fox, Sarah Bernays.

**Data curation:** Dorothy Drabarek, Duy Trinh-Hoang.

**Formal analysis:** Dorothy Drabarek, Duy Trinh-Hoang, Thu Anh Nguyen, Sarah Bernays.

**Funding acquisition:** Tho T.H. Dang, Thu Anh Nguyen, Thuong Thu Do, Binh Hoa Nguyen, Dinh Hoa Vu, Greg J. Fox, Sarah Bernays.

**Investigation:** Tho T.H. Dang, Thu Anh Nguyen, Thuong Thu Do, Binh Hoa Nguyen, Dinh Hoa Vu, Greg J. Fox, Sarah Bernays.

**Methodology:** Dorothy Drabarek, Duy Trinh-Hoang, Sarah Bernays.

**Project administration:** Dorothy Drabarek, Duy Trinh-Hoang, Manisha Yapa, Tho T.H. Dang, Hai Dang Vu, Thu Anh Nguyen, Thu Thuong Do, Greg J. Fox.

**Resources:** Manisha Yapa, Tho T.H. Dang, Hai Dang Vu, Thu Anh Nguyen, Binh Hoa Nguyen, Greg J. Fox, Sarah Bernays.

**Supervision:** Manisha Yapa, Greg J. Fox, Sarah Bernays.

**Validation:** Duy Trinh-Hoang, Hai Dang Vu.

**Visualization:** Duy Trinh-Hoang.

**Writing – original draft:** Dorothy Drabarek, Duy Trinh-Hoang, Sarah Bernays.

**Writing – review & editing:** Dorothy Drabarek, Duy Trinh-Hoang, Manisha Yapa, Thu Anh Nguyen, Greg J. Fox, Sarah Bernays.

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
