## [Decision Letter · Decision Letter 0]

10 Dec 2024

PGPH-D-24-02013

Making it fit for (everyone’s) purpose: Examining the challenges of capitalising on mHealth’s potential to deliver locally-aligned person-centred care for drug resistant tuberculosis in Vietnam

Dear Dr. Trinh,

Thank you for submitting your manuscript to PLOS Global Public Health. After careful consideration, we feel that it has merit but does not fully meet PLOS Global Public Health’s publication criteria as it currently stands. Therefore, we invite you to submit a revised version of the manuscript that addresses the points raised during the review process.

We look forward to receiving your revised manuscript.

Kind regards,

Sharmistha Mishra, M.D., Ph.D

Academic Editor

Journal Requirements:

2. Please send a completed 'Competing Interests' statement, including any COIs declared by your co-authors. If you have no competing interests to declare, please state "The authors have declared that no competing interests exist". Otherwise please declare all competing interests beginning with the statement "I have read the journal's policy and the authors of this manuscript have the following competing interests:"

3. In the online submission form, you indicated that "The data used in this study cannot be shared publicly as it violates the agreement to which the participants consented. Data will only be made available on request.". 

3. Uploaded as supplementary information.

4. Please provide an Author Summary. This should appear in your manuscript between the Abstract (if applicable) and the Introduction, and should be 150–200 words long. The aim should be to make your findings accessible to a wide audience that includes both scientists and non-scientists. Sample summaries can be found on our website under Submission Guidelines:

https://journals.plos.org/globalpublichealth/s/submission-guidelines#loc-parts-of-a-submission

5. Figures 1 and 2: Please confirm whether you drew the images / clip-art within the figure panels by hand. If you did not draw the images, please provide (a) a link to the source of the images or icons and their license / terms of use; or (b) written permission from the copyright holder to publish the images or icons under our CC-BY 4.0 license. Alternatively, you may replace the images with open source alternatives. See these open source resources you may use to replace images / clip-art:

- https://openclipart.org/

6. Figures 1 and 2 contains screenshots. We are not permitted to publish these under our CC-BY 4.0 license; websites are usually intellectual property and are copyrighted. This includes peripheral graphics of the web browser such as the [X] buttons. We ask that you please remove or replace it.

Additional Editor Comments (if provided):

Reviewers' comments:

Reviewer's Responses to Questions

**Comments to the Author**

1. Does this manuscript meet PLOS Global Public Health’s publication criteria ? Is the manuscript technically sound, and do the data support the conclusions? The manuscript must describe methodologically and ethically rigorous research with conclusions that are appropriately drawn based on the data presented.

Reviewer #1: Partly

Reviewer #2: Partly

Reviewer #3: Yes

2. Has the statistical analysis been performed appropriately and rigorously?

Reviewer #1: N/A

Reviewer #2: N/A

Reviewer #3: Yes

3. Have the authors made all data underlying the findings in their manuscript fully available (please refer to the Data Availability Statement at the start of the manuscript PDF file)?

Reviewer #1: No

Reviewer #2: No

Reviewer #3: No

4. Is the manuscript presented in an intelligible fashion and written in standard English?

Reviewer #1: Yes

Reviewer #2: Yes

Reviewer #3: Yes

5. Review Comments to the Author

Reviewer #1: I want to commend the authors for the ambitious project that they have undertaken with this manuscript. It is well-written and will provide valuable insights to the field. I hope that the following comments can be used to improve the manuscript.

I received a PDF without numbered lines, so I have listed page numbers to help locate the specific areas of the text. In the next version, please follow the journal’s submission guidelines and include continuous line numbers.

The title does not seem to perfectly align with the ambitions of the manuscript. This decision is entirely up to the authors, and I will not reject the paper for this reason, but a modified title would probably be clearer to readers.

The abstract should be revisited by the authors. The study design was insufficiently explained in the abstract. It did not make clear how this study fit within a larger clinical trial. It was unclear if this was an exploratory or explanatory qualitative study (or a combination) nested within a trial. Page 7 and the conclusion describe a “case study” which was also not mentioned. Neither the qualitative analysis method nor the development of a conceptual framework were included in the abstract.

Citation 5 should cite the most recent GTB report.

In the description of the V-SMART trial, it is unclear how people were selected to receive access to a smartphone and data packages. How was socioeconomic status assessed to determine equitable access to the technology? What proportion of individuals received data and smartphones among those who participated in the quali sample? The amount of data provided monthly was unclear along with the expectations/terms for receiving & returning a loaned phone. These financial motivators could have a large impact on participants and should not be ignored. It could be argued that among the ‘performative use’ group, their performative participation in the project was in exchange for the financial return of participation, in an even more direct way than engaging with a symbol of good care. Also, among the unproductive use group, were these people provided with smartphones from the project and then insufficiently trained on how to use them? Did they pass this new asset to their kids and grandkids to provide an additional resource to the household, but then they stopped engaging with the app and continued to use the phone? These financial motivations for participation should be explored in greater detail within the manuscript.

Additionally, it is unclear in the introduction how or why the app would provide the personal phone numbers of the HCWs to the patients. “I only give my personal number to patients who tend to have more problems with medication. But now most of the patients [with the app] can access my number, they now text me too, I have to answer… it is good to help patients contact the senior doctor directly, but it becomes too much for me to handle.’’ Was it a design flaw that provided phone numbers to participants instead of centralizing communication within the app? Did the HCWs consent to having their number distributed prior to the study?

The design section needs some clarification. What is the precise study design? Is it a qualitative study or a qualitative case study? If it is the former, then I would recommend adhering to the Consolidated criteria for reporting qualitative research- the COREQ checklist- and mentioning your methodological orientation. If it is the latter, then I would recommend grounding it in other research on case study designs. For example: Priya, A. (2021). Case Study Methodology of Qualitative Research: Key Attributes and Navigating the Conundrums in Its Application. Sociological Bulletin, 70(1), 94-110.

The aim of the paper does not appear to be consistent throughout. You wrote, “The qualitative evaluation aimed to adapt the app to better align with the needs of the trial’s end users.” There is some uncertainty about whether this paper primarily focuses on evaluating the Bac Sy Minh app to improve its use for MDR-TB or to use this local case study to develop a broader iterative approach to evaluate mHealth interventions. The development of a conceptual framework and the presentation of the results and discussion seem to aspire to the broader ambition, but this is not included in the stated aim. Please clarify.

Then you wrote, “We used individual in-depth interviews to understand…the app’s acceptability…” This manuscript does not sufficiently engage with the topic of acceptability. The only point the paper seems to explicitly make about acceptability is that the social is a mediator of acceptability. I could not discern a working definition of acceptability in the manuscript. The word acceptability does not appear in the results. Would the Bac Sy Minh app even be considered acceptable? A commonly cited theoretical framework of acceptability (Sekhon, M., Cartwright, M. & Francis, J.J. Acceptability of healthcare interventions: an overview of reviews and development of a theoretical framework. BMC Health Serv Res 17, 88 (2017).) proposes multiple constructs forming an understanding of acceptability. You do not need to accept this suggestion, but its concept of ‘intervention coherence’ may help explain how misunderstandings of the “remit of the app” lowered its acceptability and how the added burden placed on the HCWs could also diminish acceptability.

How was a sampling endpoint achieved- data saturation, information power, prior studies? My broader question is, why was purposive sampling for maximum variation stopped with only 2 patients in the functional use and four in the enhanced use groups?

Page 12- Please explain why codes were inductively developed from interview summaries and not transcripts.

Did the authors complete framework analysis? The type of qualitative analysis is unclear and pages 12 & 13 do not include any citations in the section on analysis.

The methods should describe how the conceptual framework was developed. How were participants categorized into the four use patterns?

What was done to improve trustworthiness (credibility, transferability, dependability, confirmability) in the qualitative research?

Page 18- “The focus on adopting a ‘good care’ logic shaped the adaptation of AEs and outpatient care in practice.” The phrase “adaptation of AEs” was a bit unclear. Do you mean AE reporting or how patients perceived their AEs?

Social and functional value seemed to also played a large role in the use of the app by HCWs, but this is not reflected in the conceptual framework. Their limited and tentative use in the model is only described as 'PCC logic' in Figure 2, but the paper talks about unmet expectations, misaligned functionality, the burden of engaging with the app, duplication of work, etc. Do the authors feel that the conceptual framework sufficiently describes the HCW’s use case?

How does time work in the conceptual model? The text describes a gradual decline in app usage over time, but did participants shift categories throughout care? For example, for those classified as being in the performative group, did they start in the functional or enhanced user groups and then convert to performative use by the end of treatment? As their participation changed over time, at what point were they categorized as falling into one of the quadrants? This should be addressed in the methods and results.

Page 25- The functional use group only included 2 people. Do the authors feel that this is a large enough sample to include a profile of the group?

I would suggest renaming category 4 in the framework since the adjective ‘unproductive’ could potentially also apply to the performative use category. It sounds more like the digital divide created non-use or family-enabled use.

The discussion does not engage with the literature on Human Centered Design, which addresses many of the iterative design practices for technology/mHealth that are promoted in the paper.

The conclusions should be updated to align with my earlier comments regarding the aim of the paper.

Reviewer #2: I thank the authors for documenting the use of a digital tool to improve pharmaco-vigilance in an MDR-TB Program in Vietnam. Please receive, below, a few comments for consideration:

(a) the manuscript title is not concise. Please cut out some words and include descriptions that aptly reflect the content of the manuscript such as the study design. The phrase, "Making it fit...., examining the challenges on ... potential..." needs revision.

(b) In the abstract and introduction sections, the justification for optimal adverse event monitoring in DR-TB programs is not adequately presented. In the introduction, I suggest that the authors provide a rationale for timely adverse event monitoring in the DR-TB program. Please also share statistics on the incidence of the adverse events in Vietnam, and how these have been managed. Provide insights on the WHO guidance and national policies on pharmaco-vigilance in the DR-TB program. This will make the case for use of digital technologies in these programs. What is the mobile phone penetration rate in Vietnam - is this the same across all the sampled provinces?

(c) The authors report that they observed that there was a reduction in use of the app, but there are no numbers presented to support this assertion.

(d) Did the authors use any theoretical model to guide their qualitative inquiry?

(e) In the results section, the authors discuss their findings - this should fit in the discussion section. The results section should answer the question, "What did you find?"

Reviewer #3: Intro

- The introduction is quite lengthy and could be more focused. It would benefit from streamlining the background information, emphasizing the specific gap in the literature, and presenting the study's objectives more clearly. Reducing repetition and prioritizing relevant studies will help make the introduction more concise and directly aligned with the research question.

- In the second paragraph, instead of referring to all drugs used for TB or MDR as antibiotics, it would be more accurate to use the term 'drug regimen'

Result

- describe the sample subsection in the methods section, focusing on participant characteristics in the results section. Move details unrelated to participant characteristics to the methods section.

-Why were significantly more male patients interviewed compared to females, nearly double in number? What was the reason for this discrepancy?

- The organization of the results section appears somewhat scattered and does not clearly reflect the themes and sub-themes. I recommend reorganizing the results for better clarity. You could include a table or graphic to summarize the themes and sub-themes related to the challenges faced by both healthcare workers and patients. This would provide a clear overview of the challenges at a glance and make it easier to follow the rest of the results.

Discussion

-The discussion is rich in content but could benefit from a more streamlined structure to avoid overwhelming the reader

- The narrative drifts into broader reflections on mHealth interventions in general

- Certain sentences are overly long and complex, making the content harder to follow

- Providing suggestions for future studies on how to delve deeper into the social and functional dynamics of mHealth interventions, especially within complex healthcare settings, would be valuable.

- lack of actionable insights for policymakers and implementers

6. PLOS authors have the option to publish the peer review history of their article (what does this mean? ). If published, this will include your full peer review and any attached files.

**Do you want your identity to be public for this peer review?** For information about this choice, including consent withdrawal, please see our Privacy Policy .

Reviewer #1: No

Reviewer #2: No

Reviewer #3: No

---

## [Editor Report · Decision Letter 1]

11 Mar 2025

Examining the challenges in sustaining user engagement with a mobile app to enhance multidrug-resistant tuberculosis (MDR-TB) care in Vietnam and its implications for implementing person-centred mHealth interventions

PGPH-D-24-02013R1

Dear Mr Trinh,

We are pleased to inform you that your manuscript 'Examining the challenges in sustaining user engagement with a mobile app to enhance multidrug-resistant tuberculosis (MDR-TB) care in Vietnam and its implications for implementing person-centred mHealth interventions' has been provisionally accepted for publication in PLOS Global Public Health.

Best regards,

Sharmistha Mishra, M.D., Ph.D

Academic Editor

Thank you for carefully addressing and responding to the peer-reviewer comments and suggestions, in the revised manuscript.